# Development of Alginate–Pullulan Capsules for Targeted Delivery of Herbal Dietary Supplements in Functional Fermented Milk Products

**DOI:** 10.3390/foods14162878

**Published:** 2025-08-19

**Authors:** Alibek Muratbayev, Berik Idyryshev, Aitbek Kakimov, Aigerim Bepeyeva, Madina Jumazhanova, Marzhan Tashybayeva, Gulmira Zhumadilova, Nazerke Muratzhankyzy, Zhadyra Imangaliyeva, Aray Bazanova

**Affiliations:** 1Research School of Food Engineering, Shakarim University, 20A Glinki Street, Semey 071412, Kazakhstan; 2Faculty of Food Technology, Almaty Technological University, 100 Tole bi Str., Almaty 050010, Kazakhstan

**Keywords:** encapsulation, sodium alginate, pullulan coating, rheological properties, fermented milk beverage, gastrointestinal release, microcapsule morphology

## Abstract

The present study develops and optimizes a jet-cutting encapsulation method using a laboratory-scale encapsulator to incorporate herbal dietary supplements into fermented milk products. Sodium alginate and pullulan were selected as core and coating polymers, respectively, after rheological screening demonstrated that 1% alginate (η ≈ 350–450 Pa·s at 22–25 °C) and 2% pullulan (η ≈ 400 Pa·s at 25–30 °C) provide a balance between atomization, shell integrity, and fluidity. Under optimized conditions, capsules of 1.00 ± 0.05 mm diameter and high sphericity (aspect ratio 1.08 ± 0.03) were produced. In vitro gastrointestinal simulation confirmed capsule stability in simulated gastric fluid (pH 2.0) and complete disintegration within 120 min in simulated intestinal fluid (pH 7.2). Inclusion of 8% (*w*/*w*) capsules in a fermented milk beverage preserved appearance, texture, flavor, and color while increasing viscosity from 2.0 to 4.0 Pa·s. Titratable acidity rose from 87 °T at 24 h to 119 °T at 120 h, with sensory quality remaining acceptable; substantial gas formation and excessive sourness occurred only after 168 h, defining a 5-day refrigerated shelf life. These findings demonstrate that the 1% alginate–pullulan capsule system successfully protects plant extracts during gastric transit and enables targeted intestinal release, while maintaining the sensory and rheological properties of the fortified fermented milk product.

## 1. Introduction

The encapsulation of bioactive agents involves enclosing small particles (solid, liquid, or gaseous) within a polymeric shell or matrix to form micro- or nano-scale beads. These encapsulation technologies serve to stabilize sensitive compounds, control their release, and protect them from adverse environmental conditions [1,2]. Among the various encapsulation strategies, differences in processing methods can significantly influence capsule size, structure, and functional performance, making the choice of technique critical for specific applications.

The jet-cutter encapsulation technique involves the continuous extrusion of the polymer–core mixture through a nozzle, followed by mechanical cutting of the emerging liquid jet into uniform droplets before gelation. Compared with spray drying, extrusion, and electrostatic droplet generation, this method offers several benefits. It produces uniform beads with controlled size, achieves higher throughput than dripping or electrostatic methods, and operates at mild temperatures, protecting heat-sensitive bioactives such as probiotics and polyphenols [3]. The process is adaptable to different viscosities and encapsulating materials, enabling use in various food systems. Limitations include the need for precise synchronization between feed and cutting speeds to maintain size uniformity and reduced sphericity at very high production rates. Unlike spray drying, it yields hydrated beads rather than powders, which may limit dry product applications but enhances bioactive stability in hydrated or semi-solid matrices. This balance of gentle handling, uniformity, and productivity makes the jet-cutting technique suitable for commercial functional dairy applications [4,5].

In the pharmaceutical and agrochemical industries, encapsulation has long been employed to enhance efficacy, reduce toxicity, and mask undesirable tastes or odors. More recently, its application in food science and biotechnology has expanded rapidly, enabling the development of functional foods enriched with probiotics, vitamins, antioxidants, and other health-promoting ingredients [6,7].

In food applications, encapsulation of lactic acid bacteria, probiotic strains, or herbal extracts is particularly valuable for protecting sensitive payloads from gastric acidity and ensuring targeted delivery to the intestine [8]. Polysaccharides such as pectin and sodium alginate are widely employed due to their biocompatibility, gel-forming capability, and responsiveness to pH changes. Alginate, for example, undergoes rapid ionic cross-linking in the presence of calcium ions to form hydrogels that remain intact in acidic environments but swell and dissolve under neutral to alkaline conditions. Pectin complements this behavior by providing tunable network strength and film-forming properties [9,10,11].

To better understand the benefits of encapsulating herbal extracts, it is essential to examine the pharmacological properties of their key bioactive constituents. These phytochemical compounds, including tannins, saponins, and flavonoids, not only define the therapeutic value of the encapsulated supplements but also influence their stability, bioavailability, and targeted delivery within the gastrointestinal environment [12,13,14].

Tannins, saponins, and flavonoids exhibit broad pharmacological activities, such as antioxidant and immunostimulatory effects that enhance organismal resilience to environmental stressors [15,16]. Oral bioactive compounds, however, often suffer degradation in the gastrointestinal tract, limiting their therapeutic potential. Encapsulation within biocompatible polymers, such as pH-sensitive alginate matrices, protects these compounds from acidic gastric conditions and enables controlled release in the intestine, where enterocytes lining over 90% of the surface facilitate flavonoid absorption into systemic circulation [17]. Encapsulation increases the absorption of active compounds, hides any unpleasant flavors, and keeps the compounds chemically stable. Polymers that respond to triggers such as changes in pH, the presence of digestive enzymes, or temperature can be chosen so that they break down at the right time and place, releasing the active ingredient where it is needed and maintaining its effect over a longer period [18,19].

The objective of this study was to develop and optimize a scalable encapsulation process using alginate-based capsules for herbal dietary supplements. Further, the study aimed to evaluate the capsules’ physicochemical, rheological, and sensory characteristics when added to fermented milk, ensuring targeted gastrointestinal delivery.

## 2. Materials and Methods

### 2.1. Materials

Medicinal plants (echinacea, levzeya, and rosehip) growing in the Abai and East Kazakhstan regions of the Republic of Kazakhstan were used for encapsulating dietary supplements.

Sodium alginate (C_6_H_7_O_6_Na)_n_, amidated pectin, and calcium chloride (CaCl_2_) were obtained from KazKhimBaza LLP (Almaty, Kazakhstan) and used as encapsulation materials in this study. Pullulan (CAS 9057-02-7) was purchased from Qingdao Kenco Co., Ltd. (Qingdao, China).

### 2.2. Preparation of Tincture from Medicinal Plants for Further Encapsulation

A liquid extract was prepared from a blend of Echinacea purpurea aerial parts, Levzeya officinalis root, and Rosa canina fruits, all sourced from the Abai and East Kazakhstan regions. Each botanical material was air-dried to constant weight at ambient temperature, shielded from direct sunlight, and milled to 3–5 mm particles. Equal masses (100 g each) of the three plant powders were homogenized and loaded into a stainless-steel mesh basket within a jacketed percolator. A 35% (*v*/*v*) aqueous ethanol solution—selected for its efficacy in extracting both polar and semi-polar phytoconstituents—was introduced at a plant-to-solvent ratio of 1:5 (*w*/*w*), and the system was maintained at 20–30 °C for 12–24 h under closed conditions to promote maceration. The percolate was then agitated continuously for 1 h, and the temperature was sequentially elevated in three equal increments to 50–70 °C to enhance solute diffusion, followed by a further 12–24 h infusion period. A volume of extract equivalent to the original plant charge was collected, fresh 35% ethanol was added at a 1:1 ratio, and the residue underwent a second 12 h infusion with intermittent mixing. All fractions were pooled and subjected to cold settling (5–8 °C) for five days to remove insoluble impurities. The clarified tincture was finally filtered through a cotton-gauze medium and stored in amber glass bottles to protect labile constituents from photodegradation [20].

### 2.3. Description of the Developed Encapsulation Unit Based on the Disc Spray Method

A jet-cutter encapsulator was employed to produce calcium-alginate microcapsules (Figure 1). The apparatus comprises a stainless-steel frame supporting the primary components: two 10 L feed tanks (one for the active alginate-tincture slurry and one for wash solution), a peristaltic pump, a rotating disc nozzle module, and a cylindrical reaction vessel fitted with a conical perforated separator.

The active feed, consisting of 1% (*w*/*w*) sodium-alginate solution loaded with herbal tincture, was delivered from its reservoir through silicone tubing to the peristaltic pump at a controlled flow rate (5 mL min^−1^). Upon exit from the pump, the laminar jet impinged on the edge of a rotating stainless-steel disc (4 cm diameter, 2000 rpm), which sheared the jet into uniform droplets. These droplets immediately entered the reaction vessel, where they were enveloped by a tangentially circulating 0.15 M CaCl_2_ gelling solution maintained at 22 ± 2 °C. Ionic cross-linking of the droplet surface ensued, forming a coherent alginate shell.

Formed beads were allowed to mature for 30 min under gentle recirculation, during which they rolled along the reactor wall and spiraled down the conical chamber. The perforated conical separator (pore size 0.5 mm) retained the capsules while allowing excess CaCl_2_ solution to drain. Capsules then collected in the reactor base were discharged via a bottom-mounted valve into a sterile collection vessel. Key process parameters, including feed rate, disc speed, bath temperature, and residence time, were monitored continuously and adjusted to ensure reproducible capsule size (1.0 ± 0.1 mm) and shell integrity [21].

In each encapsulation run, approximately 50 mL of polymer–bioactive solution was processed through the jet-cutter encapsulator. This volume was selected to ensure stable droplet formation and consistent capsule morphology under the chosen operating parameters. The same batch volume was used for all replicates to maintain comparability of results.

### 2.4. Technology for Producing Capsules with Dietary Supplements

To justify the selection of the encapsulating matrix, experimental trials were conducted to assess the capsule-forming capability of pectin and sodium alginate at polymer concentrations of 0.5%, 1%, 2%, and 3% (*w*/*w*). Each solution was processed under identical jet-cutting conditions, and resulting beads were evaluated for sphericity by optical microscopy. The primary aim of these experiments was to identify the concentration that yielded capsules with the highest degree of roundness as optimal spherical geometry is critical for achieving uniform shell thickness and functional performance.

All processing steps were performed under aseptic conditions to ensure product safety. Sodium-alginate (0.5–3.0% *w*/*w*) and pullulan (5% *w*/*v*) were obtained from commercial suppliers, and the herbal tincture, composed of Echinacea purpurea, Levzeya officinalis, and Rosa canina extracts, served as the active core material. A 0.15 M calcium-chloride solution was prepared as the gelling bath.

The alginate–tincture feed solution was prepared by combining sodium-alginate powder with the mixed herbal extract in a 1:1 mass ratio and stirring at ambient temperature for one minute until a homogeneous slurry formed. This premix was delivered via peristaltic pumping to a rotating spray disk within the encapsulation reactor. As the alginate jet impinged on the disk, discrete droplets were generated and immediately introduced into the circulating CaCl_2_ bath, initiating instantaneous ionic gelation of the capsule shell. Gel beads were allowed to cure for 30 min at room temperature, during which they rolled along the reactor wall and spiraled downward to a conical perforated separator that removed residual gelling solution. Beads were collected in the reactor base and transferred directly to the next step without intermediate washing.

For secondary coating, gelled beads were immersed in 5% pullulan solution maintained at 35–40 °C and stirred gently for 20–25 min to form an additional protective layer. Coated capsules were filtered through a fine mesh sieve and then dried in a convection oven at 30–50 °C until a consistent, non-tacky surface was achieved. Final product was stored at 2–6 °C or immediately incorporated into fermented milk formulations. Throughout the procedure, processing parameters (flow rate, bath temperature, and rotor speed) were monitored and adjusted to maintain uniform bead size and shell integrity.

### 2.5. Mathematical Models for Release Kinetics

Release of the dietary supplements from capsules was described using first-order kinetics and the Hixson–Crowell and Korsmeyer–Peppas models [22,23].

First-order model describes dissolution in which the release rate depends on the concentration of the active compound (Formula (1)) [24]:
(1)$$\log_{10}C_{t}=\log_{10}C_{0}-\frac{K_{1}\cdot t}{2},$$ where *C_t_*—the concentration of the active compound in the release medium at time *t*,

*C*_0_—the initial concentration at *t* = 0,

*K*_1_—the first-order release rate constant.

Hixson–Crowell cube-root law model applies when surface area and characteristic dimension change during dissolution (Formula (2)) [25]:
(2)$$W_{t}^{1/3}=W_{0}^{1/3}-K_{HC}t,$$ where *W_t_*—the amount of active compound remaining in the dosage form at time *t*,

*W*_0_—the initial amount,

*K_HC_*—the Hixson–Crowell rate constant.

Korsmeyer–Peppas model relates the fraction released to time through a power law (Formula (3)) [26]:
(3)$$\frac{M_{t}}{M_{\infty }}=Kt^{n},$$ where *M_t_*/*M_∞_*—the fraction released at time *t*,

*K*—a constant reflecting structural and geometric features of the dosage form,

*n*—the release exponent that indicates the predominant release mechanism (Table 1)

**Table 1 foods-14-02878-t001:** Interpretation of diffusion release mechanisms from the polymer layer in the Korsmeyer–Peppas mode.

Release Rate (*n*)	Mechanism of Substance Release	Time Dependence
0.45 ≤ *n*	Diffusion according to Fick’s law	*t* ^−0.5^
0.45 < *n* < 0.89	Diffusion not according to Fick’s law	*T* * ^n^ * ^−1^
*n* = 0.89	Case II transport	Zero-order release
*n* > 0.89	Anomalous diffusion (Super Case II)	*T^n^* ^−1^

### 2.6. Technology for Producing Fermented Milk Beverages

The prototype fermented milk beverages were formulated to contain 2–10% (*w*/*w*) spray-encapsulated dietary supplements, with the remaining composition comprising 85–93% whole milk (2.5% fat) and 5% starter culture (Table 2). All ingredients met national regulatory standards and underwent incoming quality control before processing.

Milk standardization and homogenization were performed in a stainless-steel continuous unit. Raw milk was adjusted to 2.5% fat, heated to 60–65 °C, and homogenized at 15–17 MPa to ensure a uniform fat globule distribution. The homogenized milk was immediately pasteurized at 80–85 °C for 2–3 min, then rapidly cooled to the fermentation temperature.

Fermentation was initiated by transferring the cooled milk (40–42 °C) to the fermentation tank and inoculating with 5% (*w*/*w*) mesophilic starter culture under gentle agitation. The mixture was held at 40–42 °C for 5.0 ± 0.5 h, during which pH decline and rheological changes were monitored to confirm gel formation. Upon reaching the target acidity, the curd was cooled to 4–6 °C.

Encapsulated supplements were dispersed into the chilled fermented milk under slow stirring for 15 min to achieve homogeneous bead distribution without damaging capsule integrity. The final product was filled aseptically into 250 mL polypropylene containers and sealed. Finished units were stored at 4–6 °C with 80–85% relative humidity and transported under refrigerated conditions. All processing parameters—including temperatures, pressures, and times—were continuously recorded to ensure reproducibility and compliance with quality specifications.

### 2.7. Viscosity Measurement

Apparent viscosity was determined using a Brookfield RVT rotational viscometer equipped with a temperature-controlled sample cup (AMETEK, Inc., Middleboro, MA, USA). Spindle No. 4 was selected for all measurements, and rotor speeds were varied to assess shear-rate dependence. Samples were equilibrated at the target temperature (10 °C or 50 °C) before measurement to reflect gelation conditions. All determinations were performed in triplicate, and results are reported as mean ± standard deviation.

### 2.8. Microscopic Imaging and Capsule Size Analysis

Images were captured using a Mikmed-6 (JCS «LOMO», St. Petersburg, Russia) with digital camera (8.3 megapixel, ULTRA HD) within 30 min of production. Mean capsule diameter was determined from measurements of at least 50 particles per batch and is reported as the arithmetic mean ± standard deviation.

### 2.9. Quantitative Analysis of Flavonoids

A 1.00 g aliquot of sample powder (particle size ≤ 1 mm) was extracted with 30 mL of 70% (*v*/*v*) ethanol in a 100 mL round-bottom flask. The mixture was heated in a boiling water bath for 30 min, then rapidly cooled under running tap water. The extract was filtered through Whatman No. 1 paper into a 100 mL volumetric flask, and the flask was rinsed with additional 70% ethanol and diluted to volume. This preparation was designated “Solution A”.

For the test reaction, 4 mL of Solution A was transferred to a 25 mL volumetric flask, mixed with 2 mL of 2% (*w*/*v*) AlCl_3_ in 95% ethanol, and diluted to the mark with 95% ethanol. The solution was allowed to stand for 20 min at ambient temperature. A blank was prepared by replacing AlCl_3_ with one drop of 0.1 N HCl, then diluting and mixing in the same manner [27].

Absorbance was measured at 410 nm using a UV spectrophotometer (Specord 210 plus, Analytik Jena, Jena, Germany) and 10 mm quartz cuvettes. The blank was used to zero the instrument. Flavonoid content (%) was calculated from the absorbance of the test solution according to the following Formula (4):
(4)$$X\;(\%)=\frac{D\times 100\times 100\times 25}{330\times 4\times m\times (100-W)},$$

*D*: Optical density of the tested solution,

330: Specific absorption index of the complex with aluminum chloride at 410 nm,

*m*: Mass of the sample (g),

*W*: Mass loss on drying (%).

### 2.10. Qualitative and Quantitative Tannin Analysis

#### 2.10.1. Qualitative Test

A 0.300 g sample of powdered plant material was suspended in 10 mL distilled water and heated in a boiling water bath for 30 min. After cooling, the mixture was filtered through Whatman No. 42 paper. To 5 mL of filtrate, 2–3 drops of 0.1% FeCl_3_ solution (Sigma-Aldrich, St. Louis, MO, USA) were added. The appearance of a brownish-green color confirmed the presence of tannins.

#### 2.10.2. Quantitative Determination

Two grams of sieved plant sample (3 mm) were extracted with 250 mL boiling water in a 500 mL Erlenmeyer flask for 30 min, with periodic stirring. The extract was cooled, reweighed, and returned to its original mass with boiled water. A coarse filtration was performed through absorbent cotton into a 250 mL flask. A 25 mL aliquot of the filtrate was transferred to a 1 L flask, diluted with 500 mL water, and acidified with 25 mL indigo-sulfonic acid solution (Merck, Darmstadt, Germany) to yield a blue solution. This solution was titrated with 0.02 M KMnO_4_ (Merck, Germany) until a faint yellow endpoint persisted for 30 s. A reagent blank, containing no sample, was titrated identically. Tannin content (mg g^−1^) was calculated from the difference between sample and blank titration volumes, sample mass, and KMnO_4_ concentration, following the procedure in [28]. All assays were performed in triplicate.

The content of tannins (X, %) converted to absolute dry raw materials was calculated using Formula (5):
(5)$$X(\%)=\frac{(V-V_{1})\times 0.004157\times 250\times 100\times 100}{M\times 25\times (100-W)},$$

V: The amount of potassium permanganate solution (0.02 mol/L) used for the titration of the extraction (mL),

V_1:_ The amount of potassium permanganate solution (0.02 mol/L), used for titration in control experiment (mL),

0.004157: Amount of tannins equivalent to 1ml of potassium permanganate solution (0.02 mol/L) per tannin (g),

M: Sample weight (g),

W: mass loss on drying (%),

250: total extraction volume (mL),

25: Extraction volume taken for titration (mL).

### 2.11. Sensory Evaluation of Fermented Milk Beverage

The sensory evaluation of the fermented milk beverage was conducted according to the recommended procedures for dairy products of varying viscosities, including both liquid and viscous fermented milk formulations [29]. Sensory evaluation was conducted using a panel of 11 trained assessors from the School of Food Engineering, Shakarim University. Each assessor independently evaluated five attributes—smell, taste, color, appearance, and consistency—using a five-point hedonic scale, where 1 indicated “poor” and 5 indicated “excellent.” The results were averaged for statistical analysis.

Samples were prepared following standard sampling protocols. For products in large containers, a minimum of 500 g was taken. For consumer-sized packaging, multiple units were selected. Prior to evaluation, samples were stored at the temperature indicated on the packaging or maintained at (4.0 ± 2.0) °C if unspecified. Sensory assessments were performed at (12.0 ± 2.0) °C.

The assessment was carried out under standardized conditions using the following equipment: thermostatic chamber, thermometer, glassware, beakers, spoons, mouth-cleansing water (30–40 °C), and individual cups for serving 50–100 g sample portions to each panelist.

Assessment Procedure: Appearance was evaluated based on color, clarity, phase separation, presence of serum, and visual impurities. Odor and aroma were assessed by direct sniffing and tasting to detect characteristic and off-odors. Texture and consistency were evaluated by stirring the product and analyzing its viscosity and uniformity in the mouth.

### 2.12. Determination of Protein, Fat, Moisture, Ash and Carbohydrates in Milk Product

Protein and fat contents of the fermented milk product were determined in accordance with national standards. Protein content was measured using the Kjeldahl method, as described in [30]. Fat content was determined according to [31]. Moisture and dry matter contents of the fermented milk product were determined in accordance with [32]. The carbohydrate content was calculated by difference, subtracting the measured amounts of protein, fat, and ash from the total composition.

### 2.13. Determination of Titratable Acidity

Titratable acidity was measured by potentiometric titration using 0.1 N sodium hydroxide (NaOH) and phenolphthalein indicator, in accordance with GOST 3624-92 [33]. A 10 mL aliquot of fermented milk was transferred to a 250 mL Erlenmeyer flask and diluted to 100 mL with distilled water. Between two and three drops of 1% (*w*/*v*) phenolphthalein solution were added as the endpoint indicator. The mixture was stirred continuously with a magnetic stirrer while titrating with 0.1 N NaOH until a persistent light pink color appeared, remaining stable for at least 1 min. The volume of NaOH consumed (V, mL) was recorded.

Titratable acidity (°T) was calculated as the volume of 0.1 N NaOH required to neutralize the acids present in 100 mL of product (Formula (6)):
(6)$$TA=V\cdot \frac{100}{V_{s}}$$ where *TA*—titratable acidity, °T;

*V_s_*—the volume of fermented milk sample, 10 mL;

*V*—volume of NaOH, mL.

All determinations were performed in triplicate, and results are reported as mean ± standard deviation.

### 2.14. Statistical Analysis

All experiments were conducted in triplicate, and data are presented as mean ± standard deviation. Statistical comparisons among multiple groups were performed using one-way analysis of variance (ANOVA) followed by Tukey’s post-hoc test to identify significant pairwise differences. A significance threshold of *p* < 0.05 was applied.

## 3. Results

### 3.1. Study of Capsule Shapes and Sizes

At 0.5–2% pectin, droplets gelled only partially on contact with the CaCl_2_ bath, yielding deformable beads with irregular outlines and a rough, porous surface. Increasing the concentration to 3% exacerbated jet breakup due to the high viscosity of the feed (Figure 2). The resulting agglomerates exhibited a broad size distribution (1.8 ± 0.4 mm) and failed to form discrete capsules. All pectin formulations failed to produce capsules that were both nearly spherical (aspect ratio ≤ 1.1) and mechanically stable when stirred. Pectin failed to form stable, spherical capsules because its chain structure and solution conditions did not favor rapid, uniform gelation in calcium chloride. High esterification or acetylation blocks carboxyl groups needed for Ca^2+^ “egg-box” cross-links, while branched regions sterically hinder network growth. At 0.5–2% pectin, the weak, patchy skin that develops collapses into irregular beads; at 3%, the feed becomes too viscous for droplet breakup, resulting in filaments and agglomerates. Additional solutes—monovalent salts, polyphenols, or proteins—can further compete for Ca^2+^ or distort the interface, reducing cross-link density and preventing coherent capsule walls [34,35].

Alginate solutions exhibited a markedly different behavior. A 0.5% formulation produced highly spherical beads (aspect ratio 1.03 ± 0.02) with a smooth, glossy surface and a mean diameter of 1.00 ± 0.05 mm (Figure 3). Raising the polymer level to 1% preserved surface homogeneity but increased average diameter to 1.10 ± 0.06 mm and caused a slight loss of circularity (aspect ratio 1.08 ± 0.03), presumably due to faster outer-shell solidification that impeded droplet rounding. Further increases (1.5%) led to elongated or teardrop-shaped particles (1.40 ± 0.08 mm) with visible surface striation, while 2% and 3% alginate solutions could not be sprayed, producing viscous threads that collapsed into fragments rather than capsules. These observations confirm that excessive viscosity hampers droplet breakup in the spray field.

Alginate forms capsules with ease. Its guluronate blocks carry closely spaced carboxyl groups that bind Ca^2+^ rapidly, creating an “egg-box” gel that solidifies the droplet surface before any deformation occurs [36]. At 0.5%, the solution viscosity remains within the capillary-instability window, so the jet breaks into uniform droplets that round under surface tension, creating smooth, spherical beads. At 1% alginate, the surface gels more quickly. The outer layer solidifies while the interior is still fluid, which stops the droplet from fully rounding and results in slightly larger beads. Beyond 1.5%, molecular entanglement raises viscosity and elastic recoil, preventing clean jet pinching; the outgoing stream stretches into filaments that fold into elongated or fragmented masses. Thus, the balance between rapid ionic cross-linking and manageable feed viscosity explains alginate’s superior but concentration-sensitive capsule formation.

The smaller the capsule, the better its consumer properties. Consequently, 1% alginate was chosen as the encapsulating matrix for all further studies (Figure 4). To enhance barrier properties against whey migration, the beads were additionally coated with a thin pullulan layer by dip-coating in 3% pullulan solution, followed by air drying (5 min, 25 °C). Sodium alginate at an intermediate concentration (1%) provides the optimal balance between processability, sphericity, and mechanical robustness for spray-formed capsules intended for fermented dairy applications.

Published studies report a broad range of bead sizes that depend mainly on the droplet-formation method and on solution viscosity. Electrostatic or syringe extrusion typically gives sub-millimeter alginate beads: about 80–300 µm for alginate–psyllium, 200–1000 µm for alginate–fenugreek [37], around 600 µm for calcium-alginate [38], and 1.8–1.9 mm when chitosan-coated alginate is produced with larger nozzles [39]. Spray-drying used for polyphenols yields powders rather than hydrogel beads [40]. Nanoformulations produce 40–70 nm carriers that are texturally imperceptible [41]. Internal-gelation emulsions usually form micro- to sub-millimeter beads with broader size dispersion [42]. The present jet-cutting method produced smooth, spherical alginate capsules of 1.00–1.10 mm, larger than most electrostatic or extrusion beads but smaller and more uniform than pectin agglomerates (about 1.8 mm). Differences reflect the droplet-generation method, nozzle diameter, and solution viscosity.

### 3.2. Study of the Viscosity of Encapsulation Solutions

Viscosity, or internal friction, is a fundamental transport property of polymer solutions, reflecting their resistance to relative motion between adjacent fluid layers. This resistance dissipates the work applied to induce flow as heat. Achieving capsules with uniformly thick walls requires feed solutions of specific, well-defined viscosity. Viscosity is influenced by numerous factors, including solution composition, polymer concentration, temperature, pH, electrolyte content, and processing conditions [43,44]. Therefore, to formulate a solution with optimal performance characteristics, it is essential to characterize the rheological behavior of each component in aqueous media. In particular, understanding how the viscosity of a gel-forming mixture varies with polymer concentration and temperature is critical for predicting gel properties and controlling capsule formation. Accordingly, the objective of this study was to investigate the rheological properties of selected natural polymers that show promise for producing food-grade capsules.

Across the biopolymer systems examined, viscosity increased with polymer concentration but exhibited distinct magnitudes and sensitivities to temperature and shear. Sodium-alginate solutions ranged from 410 Pa·s at 0.5% to 3433 Pa·s at 3% when measured at 20 °C and 20 rpm (Figure 5), reflecting a transition from dilute coils to an entangled network above 1%. Heating to 50 °C reduced alginate viscosities by approximately 45% at low concentrations and by 60% at 2%, while increasing shear from 20 to 50 rpm produced a further 28–32% decrease, demonstrating moderate shear-thinning (Figure 6). There is a pronounced temperature dependence; at higher temperatures, the viscosity of the mixture decreases, which is associated with the destruction of the gel structure and a decrease in intermolecular bonds in the system. Visually, the differences between the temperature curves become particularly noticeable at high alginate concentrations, in which the temperature effect is most pronounced.

Pectin solutions were substantially less viscous, varying nearly linearly from 19 Pa·s (0.5%) to 160 Pa·s (3%) under the same baseline conditions. A uniform 52% viscosity drop occurred upon heating to 50 °C at all concentrations, and shear-rate increases reduced viscosity by 21–29% for 0.5–2% pectin but by only 6% at 3%, indicating early network formation (Figure 7 and Figure 8).

Pullulan displayed an intermediate profile: viscosities rose non-linearly, from 205 Pa·s at 0.5% to 1735 Pa·s at 3% at 20 °C/20 rpm, with heating to 50 °C causing a 44% reduction at low solids and up to 70% at 3% (Figure 9 and Figure 10). Shear increases lowered pullulan viscosities by 23–31% for 0.5–2% solutions and by 16% at 3%, consistent with partial chain alignment and weak gel behavior.

These rheological differences define each polymer’s processing envelope. Pectin’s low viscosity and high temperature sensitivity preclude stable spray-droplet formation, as insufficient viscous damping leads to irregular beads. Pullulan at 2% and 25–30 °C yields viscosities near 400 Pa·s, suitable for uniform dip-coating, but risks uneven films at higher solids. Only alginate achieves the balance required for spray cutting: a 1% solution at ambient conditions (η ≈ 350–450 Pa·s) supports both efficient jet breakup and robust capsule wall formation, while its shear-thinning behavior facilitates pumping without compromising bead integrity.

The data obtained shows that an increase in temperature leads to a significant decrease in viscosity, which is explained by the destruction of the gel structure and a decrease in intermolecular interactions. Temperature also plays a significant role in this relationship. As the temperature rises, viscosity may decrease due to increased molecular mobility, which leads to the breakdown of some intermolecular interactions [45]. Thus, temperature and concentration are key factors determining the rheological properties of gel-forming mixtures.

As a result of experimental studies, it has been established that, to obtain encapsulated dietary supplements, it is advisable to use polymer—1% alginate with pullulan—which provides additional protection for the capsules from the aggressive conditions of the gastrointestinal tract. Based on the experimental data obtained, a technology for obtaining capsules was developed. The produced capsules (Figure 11) exhibited a uniform spherical shape and maintained an elastic consistency that preserved their structural integrity during handling. Sensory assessment revealed no detectable taste or odor, and their color was characterized as white with a slight creamy tint. These organoleptic attributes indicate that the encapsulation process yielded beads with excellent visual appeal, textural resilience, and sensory neutrality, meeting the desired quality criteria for incorporation into food matrices.

The viscosity trends reported here are consistent with prior studies on hydrocolloids used for encapsulation. The study [37] noted that raising alginate from 1% to 3% markedly increased viscosity. This improved bead integrity, although it reduced processability in extrusion. The work [39] reported that alginate above 2% produced larger, less uniform beads because of high feed viscosity. Lower viscosities helped droplet breakup but weakened bead strength. The observations [38,42] found that 0.5–1.5% alginate provided a practical balance between flow and acid resistance in dairy systems. Another study [40] observed similar behavior for pectin and gum arabic, whose lower viscosities suited spray-drying rather than bead formation. The work [41] used nano-encapsulation with particle sizes outside the present hydrodynamic range, yet still selected polymer levels that balanced coating viscosity with efficiency. Overall, viscosity depends mainly on polymer concentration and temperature. Higher solids promote overlap of polymer chains and resistance to flow. Heating lowers viscosity by weakening hydrogen bonding. A 1% alginate feed with pullulan provided a suitable window of about 350–450 Pa·s at 20 °C for jet-cutting bead formation, giving good bead integrity without the excessive thickening seen at 2% or more.

### 3.3. Study of the Physical and Chemical Properties of Medicinal Plants

The three plants differed markedly in their phenolic profiles, with implications for both antioxidant activity and interactions with the alginate matrix during encapsulation (Table 3).

Echinacea contained 1.82% total polyphenols and 1.06% flavonoids, but exhibited the highest overall tannin levels (16.46% hydrolyzable, 3.62% condensed). The elevated tannin fraction suggests strong protein-binding capacity, which may enhance the mechanical stability of the capsule wall through secondary cross-linking with calcium ions [46]. However, excessive tannin could also lead to brittle beads or premature gelation in the feed solution.

Rosehip showed a markedly higher phenolic load, with 4.47% polyphenols and 2.75% flavonoids, yet relatively low tannin (2.70% hydrolyzable, 0.83% condensed). The abundant flavonoid content indicates potent radical scavenging activity [47], while modest tannin levels minimize the risk of feed viscosity spikes or nozzle clogging. Consequently, rosehip may deliver superior bioactivity with fewer processing challenges.

Levzeya exhibited the highest polyphenol concentration (5.28%) but a low flavonoid proportion (0.94%), accompanied by moderate tannin contents (4.41% hydrolyzable, 2.58% condensed). The disparity between total phenolics and flavonoids suggests a predominance of phenolic acids, which typically confer good water solubility and may facilitate uniform distribution within the alginate network.

The results of the study confirm that medicinal plants such as echinacea, levzeya, and rose hips are rich sources of biologically active substances that can be used to enrich fermented milk products. The high content of polyphenols, flavonoids, and tannins in the studied plants indicates their potential health benefits, including antioxidant, immunomodulatory, and anti-inflammatory effects.

Of particular interest is rosehip, which showed the highest content of polyphenols and flavonoids. These compounds are known for their antioxidant properties, which can help reduce oxidative stress and strengthen the immune system [48]. Levzeya and echinacea, in turn, are notable for their high tannin content, which makes them promising for use in products aimed at improving energy metabolism and increasing the body’s resistance to adverse external factors.

### 3.4. Study of Plant Tinctures

The resulting tincture exhibited desirable organoleptic and physicochemical characteristics consistent with quality herbal extracts. Visually, the preparation was a homogeneous, transparent liquid, free of suspended particulates or sediment. Its aroma retained the distinctive botanical notes of the source materials, while the flavor was smooth and pleasantly herbaceous without any off-tastes. The tincture presented a uniform light-brown hue, indicative of efficient pigment extraction (Table 4).

Quantitative analysis confirmed the presence of key bioactive compounds at therapeutically relevant levels: total flavonoids measured 30 mg per 100 g of tincture, tannins 21 mg per 100 g, and polyphenols 45 mg per 100 g (Figure 12). Ascorbic acid (vitamin C) content reached 62 mg per 100 g, underscoring the extract’s potential antioxidant and immunostimulatory efficacy. These findings demonstrate that the chosen extraction protocol effectively concentrates multiple classes of phytochemicals while preserving the tincture’s sensory attributes, thereby supporting its suitability for subsequent encapsulation and incorporation into functional fermented dairy products.

**Table 4 foods-14-02878-t004:** Organoleptic indicators of tincture.

Indicator	Characteristics
Consistency and appearance	Liquid, homogeneous, transparent, without sediment
Taste and smell	Pleasant taste and characteristic plant aroma
Color	Light brown, uniform throughout the mass

Thus, based on the studies conducted, a composition of echinacea, levzeya, and rosehip was selected in an optimal ratio of 1:1:1, respectively. Due to its content of biologically active substances, the resulting tincture can be used in the production of fermented milk drinks as a preventive ingredient with immunomodulatory action.

### 3.5. Study of Dietary Supplement Release in a Simulated Artificial Gastrointestinal Tract Environment

The encapsulation technology was employed to enhance the bioavailability of novel dietary supplements, given that the harsh conditions of the gastrointestinal tract can adversely affect the stability of biologically active compounds. Flavonoid absorption occurs predominantly in the small intestine, where enterocytes of the brush-border epithelium line over 90% of the luminal surface area. From this site, flavonoids and other bioactives enter systemic circulation [49,50]. Because the majority of dietary supplements are absorbed within the small intestine, the in vitro model was designed to reproduce the physiological conditions specific to this region. The choice of small intestinal simulation is justified by the highly developed microvilli system, extensive surface area, and rich enzymatic activity in this segment of the gut, which together facilitate efficient hydrolysis, transport, and uptake of nutrients and bioactive components. Modeling these conditions allows for a more accurate assessment of the release rate, extent of liberation, and potential bioavailability of the encapsulated supplement during oral administration.

Capsules containing the herbal dietary supplement were subjected to sequential incubation in simulated gastric fluid (SGF; pH 2.0) and simulated intestinal fluid (SIF; pH 7.2) in order to mimic human gastrointestinal transit. In each stage, 50 mL of the appropriate medium was pre-warmed to 37 °C and agitated at 50 rpm in a thermostatted orbital shaker. For the gastric phase, beads were immersed in SGF for 2 h, with aliquots withdrawn at 60 and 120 min to assess capsule integrity. Following this period, the remaining beads were transferred directly into fresh SIF and incubated for an additional 3 h under identical conditions; samples were taken at 60, 120, and 180 min to quantify supplement release (Figure 13).

Throughout the SGF incubation, analytical observations confirmed that the capsules remained intact, indicating effective acid resistance at pH 2.0. Upon transfer to SIF, the onset of shell dissolution was detected at 60 min, and complete disintegration of the alginate–pullulan matrix occurred by 120 min. These findings demonstrate that jet-formed 1% alginate capsules with a pullulan coating successfully protect their cargo in the gastric environment and achieve rapid release under intestinal conditions, thereby satisfying the design criteria for targeted delivery of botanical actives.

In simulated gastric fluid at pH 2.0, the capsules remained intact because the acidic conditions neutralized the alginate carboxyl groups and the calcium cross-links tightened the gel pores, preventing acid from reaching the core [51]. The external pullulan film, which is minimally soluble in acidic media, provides an additional diffusion barrier and further stabilizes the structure. After transfer to simulated intestinal fluid (SIF, pH 7.2), deprotonation of alginate restores electrostatic repulsion along the polymer backbone, while phosphate ions and bile salts compete for Ca^2+^, displacing it from junction zones [52]. These processes swell and weaken the gel; pullulan dissolves readily at neutral pH, exposing the core, and complete disintegration of the matrix occurs within 120 min. This pH-triggered sequence explains the observed acid resistance in SGF and the rapid, controlled release in SIF, confirming that the 1% alginate–pullulan capsules meet the requirements for targeted intestinal delivery of botanical actives.

Comparable findings have been reported for alginate-based capsules in simulated gastrointestinal media. Findings [37] showed that alginate and alginate–herbal gum beads (80–1000 µm) protected probiotics at pH 1.8 and released them within about 2 h under neutral or alkaline conditions. The study [39] found that chitosan-coated alginate beads (~1.9 mm) retained cells for 2 h in SGF at pH 1.55 and released contents in SIF at pH 7.4 over about 2.5 h. The study by [38] reported similar behavior for 0.6 mm alginate beads at pH 2–3 and intestinal pH. The work [42] observed improved survival of *Bifidobacterium bifidum* at pH 2.0 and release at pH 7.5 using internal gelation.

The same mechanism explains these results. Acid conditions tighten alginate gel pores. Near-neutral pH increases swelling and promotes release. Differences in bead size, coating material, and preparation route account for small variations in timing. The present study extends this evidence by showing that spray-formed alginate–pullulan capsules of 0.5–1.0 mm carrying plant extracts achieve small-intestinal release within 120 min and remain sensorially acceptable at 8% in fermented milk.

### 3.6. In Vitro Release Kinetics of Encapsulated Dietary Supplements

Release of the botanical actives from beads was quantified and fitted to standard kinetic models. Cumulative profiles showed distinct ranks in release rate: pectin > plain alginate > alginate-chitosan > alginate-pullulan. For plain alginate, 50% release occurred at ~70–80 min and ~90% at ~135 min (84% at 120 min; 92% at 140 min). Chitosan-coated beads reached ~50% at ~125–130 min and ~95% at ~170 min. Pullulan-coated beads reached ~50% at ~135–140 min and ~95% at ~180 min. Pectin beads released fastest (≈50% at ~55–60 min; ≈90% at ~115–120 min). These profiles are plotted in Figure 14 and Figure 15.

Model fitting (Table 5) identified different controlling mechanisms across formulations. For pectin and plain alginate, the Hixson–Crowell model provided the best description (highest R^2^), consistent with surface-area change and bead size reduction during release. For coated beads, the Korsmeyer–Peppas and first-order models fitted best, indicating diffusion through the coating coupled with gradual gel relaxation. The Peppas exponent nnn lay between 0.51 and 0.69 for all systems, which is characteristic of non-Fickian transport in which both diffusion and polymer erosion/swelling contribute. The slower release from coated beads reflects an added mass-transfer barrier: electrostatic complexation at the alginate–chitosan interface and the hydrophilic pullulan film both reduce permeability and extend the time to reach 90–95% release. Overall, the kinetics confirm gastric protection with moderated intestinal release and demonstrate that a 1% alginate core with a pullulan over-layer provides the most sustained profile among the tested food-grade constructs.

### 3.7. Studying the Effect of Encapsulated Dietary Supplements on the Sensory and Viscosity of Sour Milk Product

#### Sensory Evaluation

The incorporation of alginate beads containing herbal dietary supplements at levels from 2% to 10% (*w*/*w*) into a kefir-type fermented milk matrix had a dose-dependent effect on sensory attributes. Across all experimental variants (2%, 4%, 6%, 8%, and 10% bead addition), panelists rated the aroma uniformly at the maximum score (5 points), indicating that the encapsulated botanicals imparted no perceptible off-odors (Table 6). Color intensity increased markedly upon bead addition: whereas the control received an average of 4 points, all fortified samples scored 5 points, reflecting how the light-brown tint contributed by the encapsulated extracts did not detract from visual appeal.

Taste evaluation remained indistinguishable from the control (5 points) at low supplementation levels (2% and 4%), but declined to 4 points at higher loads (6%, 8%, and 10%). This modest reduction likely arises from the gradual release of tannin-rich compounds at the bead interface, introducing a subtle astringency not present in the plain product. Appearance and mouthfeel were preserved at 2–8% addition (5 points for both attributes), but at 10%, the scores for appearance and consistency fell to 4 points, suggesting that excessive bead loading may slightly disrupt the homogeneity and body of the fermented matrix.

These findings demonstrate that supplementation up to 8% (*w*/*w*) achieves immunomodulatory enrichment without compromising key quality parameters. The maintenance of texture and appearance at intermediate dosing can be attributed to the mechanical resilience of the alginate shell, which prevents premature collapse during mixing and storage. Conversely, the observed taste and consistency changes at 10% indicate a threshold beyond which particulate overload and gradual core release begin to affect product integrity. Therefore, an optimal bead concentration of 6–8% offers a balanced compromise, delivering enhanced bioactive content while retaining the sensory characteristics expected by consumers of traditional kefir-style products.

Similar effects of encapsulated bioactive addition on sensory and rheological properties have been reported. Investigations by [38] showed that adding calcium alginate probiotic beads at 0.5–2% did not affect aroma, taste, or appearance. Levels above 4% slightly increased viscosity and changed mouthfeel. The study [37] found that alginate–gum capsules with herbal extracts maintained acceptable sensory scores, with slight bitterness at higher amounts due to phenolic release. According to [40], encapsulated carotenoids deepened color while preserving flavor. Research [39] observed viscosity increases in yogurt containing probiotic beads due to interactions between particles and the gel network. The present results are consistent with these trends: 6–8% inclusion preserved flavor, appearance, and texture, whereas 10% caused mild bitterness and small textural changes. Differences in the effective levels among studies likely reflect bead size, capsule composition, and the initial viscosity of the dairy matrix.

### 3.8. Determination of the Viscosity of a Fermented Milk Beverage with Encapsulated Dietary Supplements

The incorporation of encapsulated supplements produced a clear, dose-dependent increase in apparent viscosity of the fermented milk beverage. The control sample exhibited a viscosity of 2.0 Pa·s under the defined shear conditions. Addition of 2% and 4% beads raised viscosity modestly to 2.2 and 2.5 Pa·s, respectively, reflecting the discrete particle contribution to flow resistance. At 6% loading, viscosity increased more substantially to 3.0 Pa·s, indicating the onset of particle–matrix interactions that reinforce the protein network. Further enrichment to 8% and 10% produced pronounced viscosity gains (4.0 and 7.0 Pa·s), consistent with a transition toward a percolated suspension in which densely packed beads hinder fluid deformation (Figure 16).

These rheological changes mirror the structural-mechanical observations from the organoleptic analysis. Low to moderate bead concentrations (≤6%) achieve textural enhancement without compromising pourability, whereas higher loadings (>8%) risk excessive thickening and mouthfeel heaviness. The pronounced viscosity at 10% underscores the critical balance between functional enrichment and consumer-acceptable consistency. Mechanistically, the rigid alginate–pullulan shells act as reinforcing fillers within the lactic acid gel matrix, increasing the effective solids content and promoting hydrodynamic collisions that dissipate shear energy. From a product-development standpoint, bead levels of 6–8% are recommended to maximize bioactive delivery while maintaining desirable viscosity and sensory quality.

The combined sensory and rheological data indicate that increasing the proportion of encapsulated supplements produces a proportional rise in product viscosity. At an 8% bead concentration, the fermented milk exhibits a desirable viscous consistency with uniformly dispersed 1 mm capsules, which remain imperceptible during consumption. Organoleptic assessment confirmed that this level of fortification preserves the characteristic clean, pleasantly acidic flavor and aroma of traditional fermented milk, as well as a homogeneous white appearance and smooth mouthfeel without off-notes. Furthermore, the inclusion of 8% capsules does not appreciably alter the base physicochemical parameters, since the beads remain intact and prevent premature diffusion of active compounds. Taken together, these findings identify 8% encapsulate loading as the optimal dose for delivering dietary supplements in a fermented milk matrix without compromising sensory quality or structural integrity.

Following selection of the 8% encapsulate loading as optimal, the fortified fermented milk beverage was evaluated for both sensory quality and basic compositional parameters.

The product displayed a uniform, viscous appearance, with discrete spherical beads (diameters 0.5–1.0 mm) evenly suspended throughout the matrix, confirming successful bead dispersion without phase separation. Panelists unanimously described the taste and aroma as clean, characteristic of traditional fermented milk, with no perceptible off-flavors or foreign odors. The color remained a bright, homogeneous white, and the mouthfeel was rated as smooth despite the presence of microcapsules. These observations confirm that the alginate–pullulan shells effectively mask the botanical core until ingestion, and that bead inclusion at this level does not detract from consumer-relevant sensory attributes.

### 3.9. Studying the Physicochemical Composition of Sour Milk Product

Proximate analysis of the 8%-fortified product (per 100 g) yielded 2.5 g fat, 2.9 g protein, 4.0 g carbohydrates, 0.73 g ash, and 89.87 g water, corresponding to an energy density of 53 kcal (223 kJ). These values closely match those of the control formulation (2.5% fat, 3.0% protein, 4.2% carbohydrates), indicating that capsule incorporation does not significantly alter macronutrient balance. The slight reduction in total solids reflects the displacement of milk serum by the beads, but it remains within acceptable bounds for kefir-type drinks.

Together, the sensory and compositional data demonstrate that 8% encapsulation achieves the intended functional enrichment, i.e., delivering targeted doses of herbal actives while preserving the hallmark qualities of the fermented milk product. The intact bead morphology in the chilled matrix ensures minimal premature release, thereby safeguarding both texture and flavor. Moreover, the negligible shift in proximate composition confirms that the added capsules function as inert fillers until gastrointestinal transit, supporting the feasibility of this approach for commercial development of immunomodulatory dairy beverages.

Comparable studies show that adding alginate or composite biopolymer capsules to dairy products causes little change in proximate composition. For example, yogurt with alginate-encapsulated probiotics up to 2% maintained fat, protein, and carbohydrate contents similar to the control, with only small shifts in total solids, as described in [38]. The study [42] found no significant macronutrient changes in milk drinks with encapsulated micronutrients. Small decreases in total solids were explained by the displacement of serum by the capsules, which is consistent with the present results. It was also reported by [37] that fermented milk with encapsulated herbal extracts preserved the nutritional profile while adding functional value. These studies and the present data indicate that properly dosed capsule addition does not compromise the nutritional composition of fermented dairy, supporting its suitability for functional beverage development.

### 3.10. Studying the Titratable Acidity of Fermented Milk Beverage

During refrigerated storage (4–6 °C) of the fortified fermented milk beverage, titratable acidity rose steadily from 87 °T at 24 h to 130 °T at 168 h (7 days), reflecting ongoing lactic acid fermentation by residual starter organisms. Acidity values of 87 °T and 98 °T at 24 and 72 h, respectively, remained within the typical freshness range for kefir-type drinks, whereas the increase to 119 °T by 120 h approached the upper sensory acceptance limit. By 168 h, acidity reached 130 °T, a level commonly associated with perceptible overacidification (Table 7).

Organoleptic monitoring corroborated these chemical trends (Table 8). The product maintained a homogeneous curd structure without detectable gas evolution through 72 h. At 120 h, a slight CO_2_ release was observed, although curd integrity remained intact, and aroma and flavor remained clean and pleasantly sour. By 168 h, pronounced gas formation led to a non-homogeneous texture, and panelists reported an excessively sharp sourness with a slight yeasty aftertaste, signaling the end of the acceptable shelf life under the applied storage conditions. Color remained uniformly milky white throughout, indicating that pigment stability was unaffected by acidification.

These results demonstrate that the encapsulated-supplement formulation sustains quality characteristics comparable to conventional kefir for up to five days of refrigeration, with significant quality degradation—both in acidity and sensory attributes—manifesting between 120 and 168 h. The gradual acid accumulation and eventual gas production underscore the need to define a 120 h shelf-life for the product to ensure optimal organoleptic and structural properties.

Other studies report similar acidification during the storage of fortified fermented dairy. The gradual increases in titratable acidity in probiotic yogurt during refrigeration were observed in [38]. Sensory acceptance declined when acidity exceeded about 120 °T, which matches the present limit. The work [42] showed that adding encapsulated micronutrients did not change the acidification rate relative to controls. This indicates that beads do not interfere with starter culture activity. Another study [37] found normal acid development during the first days in kefir with encapsulated plant extracts, followed by overacidification after five days and loss of flavor quality. Overall, these findings indicate that the encapsulation material preserves typical fermentation kinetics, and a five-day refrigerated shelf life is consistent with quality limits reported for related fortified dairy products.

### 3.11. Quantity of Lactic Acid Microorganisms During Storage

The total counts of lactic acid microorganisms (CFU/g) in both the control and experimental dairy products were monitored over 120 h (Table 9). The counts of lactic acid microorganisms in both the control and experimental dairy products remained stable during the first 48 h of storage, reaching log 7.18 CFU/g. A gradual decline was observed thereafter, with values in the control decreasing to log 7.00 CFU/g by day 5, while the experimental sample containing encapsulated dietary supplements retained slightly higher counts of log 7.04 CFU/g. The reduction in viable cells after 72 h is likely associated with the accumulation of metabolic products such as lactic and acetic acids, which lower the pH and create less favorable conditions for bacterial survival, as well as depletion of available nutrients and possible autolysis. The presence of encapsulated herbal dietary supplements did not suppress lactic acid bacteria and may have slightly slowed the decline in viability, likely due to the encapsulation barrier limiting direct antimicrobial effects of bioactive compounds.

This indicates that the addition of encapsulated dietary supplements did not affect the initial activity of the starter cultures. Over time, the dynamics did not change significantly and remained within acceptable limits. The differences between the samples are minimal, which indicates that encapsulated dietary supplements do not have a negative effect on the fermentation process and the vital activity of lactic acid microorganisms.

## 4. Conclusions

Alginate–pullulan microcapsules were successfully developed for targeted intestinal delivery of herbal dietary supplements in fermented milk products. Among tested polysaccharides, 1% sodium alginate formed optimal capsules, characterized by uniform spherical shape, mechanical stability, and controlled dissolution in simulated intestinal fluid. Pullulan coating provided additional protection against gastric acidity, ensuring capsule stability at pH 2.0 and complete release at pH 7.2. Incorporation of 8% capsules into kefir-type fermented milk resulted in acceptable sensory qualities, including pleasant flavor, uniform appearance, and improved viscosity (4 Pa·s). Storage studies showed product stability for five days, after which excessive acidity and gas formation indicated quality deterioration. In the encapsulated sample, viable lactic acid microorganism counts remained stable at log 7.18 CFU/g for the first 48 h, then gradually declined to log 7.04 CFU/g by 120 h, indicating that encapsulated supplements did not inhibit bacterial growth and may have slowed the loss of viability. Thus, encapsulation using alginate–pullulan represents an effective technology to protect bioactive compounds, enhance product functionality, and maintain consumer acceptance in fermented dairy applications.

## 5. Patents

Patent for a utility model of the Republic of Kazakhstan No. 9093 dated 3 May 2024. Installation for the production of encapsulated products.

## Figures and Tables

**Figure 1 foods-14-02878-f001:**
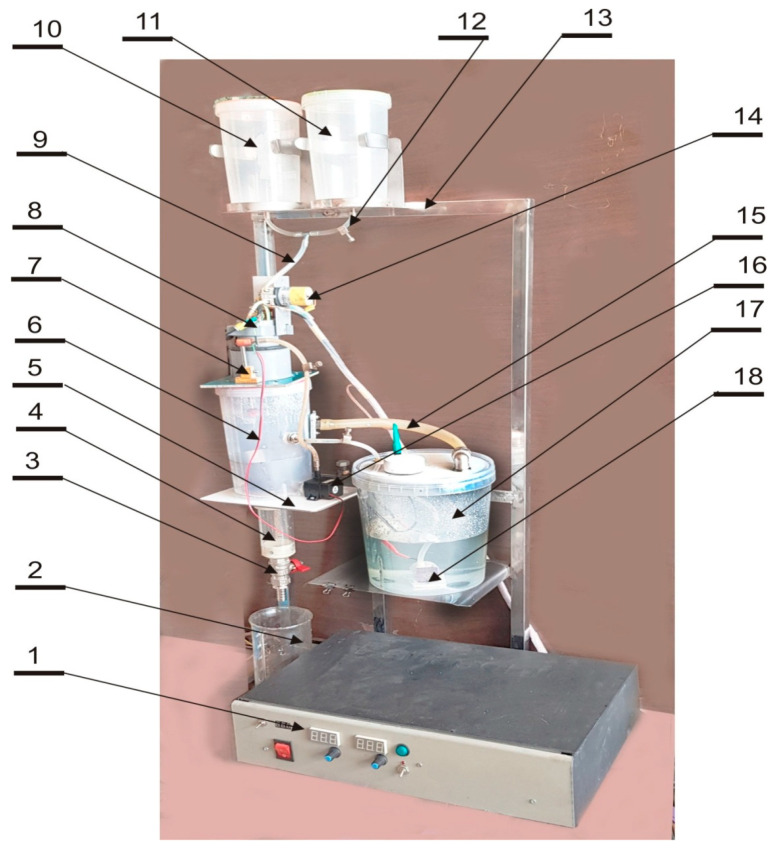
A jet-cutter encapsulator. 1—power and control unit; 2—a container for ready capsules; 3—tap of capsules release from the collector; 4—capsules collector; 5—attachment platform for the reactor and the internal circulation pump; 6—reactor; 7—nozzle for washing solution; 8—mounting frame for the electric motor of the rotor drive 9—product feed hose to the peristaltic pump; 10—container with the product; 11—container with washing solution; 12—hose clamp; 13—frame; 14—peristaltic product feed pump; 15—solution drain hose; 16—internal circulation pump; 17—supplementary container; and 18—external immersion circulating pump.

**Figure 2 foods-14-02878-f002:**
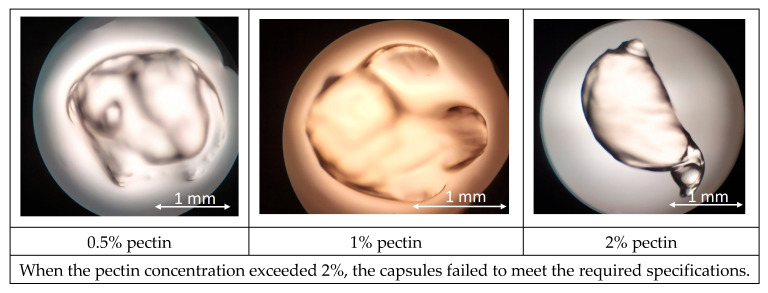
Results of microscopic observation of pectin capsules.

**Figure 3 foods-14-02878-f003:**
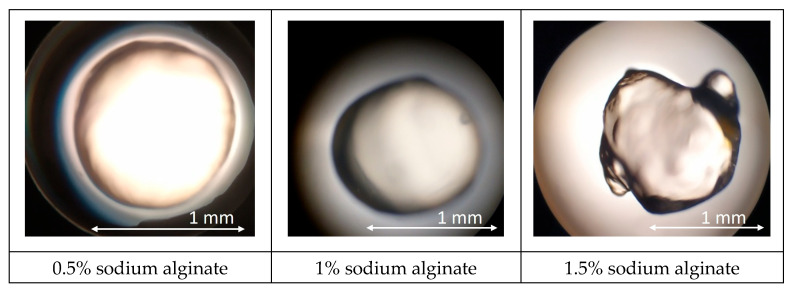
Results of microscopic observation of sodium alginate capsules (When the sodium alginate concentration exceeded 2%, the capsules failed to meet the required specifications).

**Figure 4 foods-14-02878-f004:**
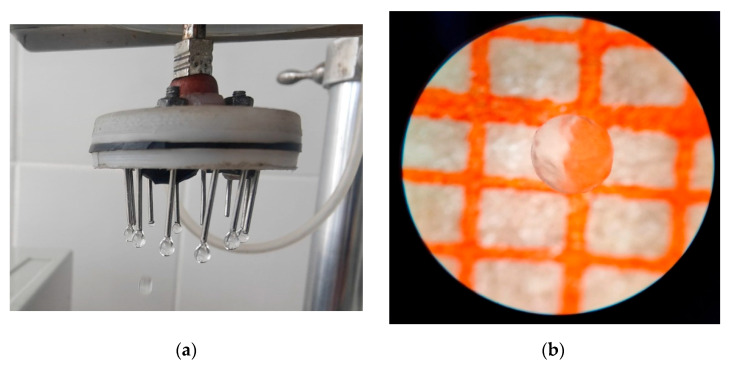
Capsules with 1% aqueous sodium alginate solution, (**a**) capsules at the outlet of the injector of the encapsulator, (**b**) view of the capsule on millimeter paper.

**Figure 5 foods-14-02878-f005:**
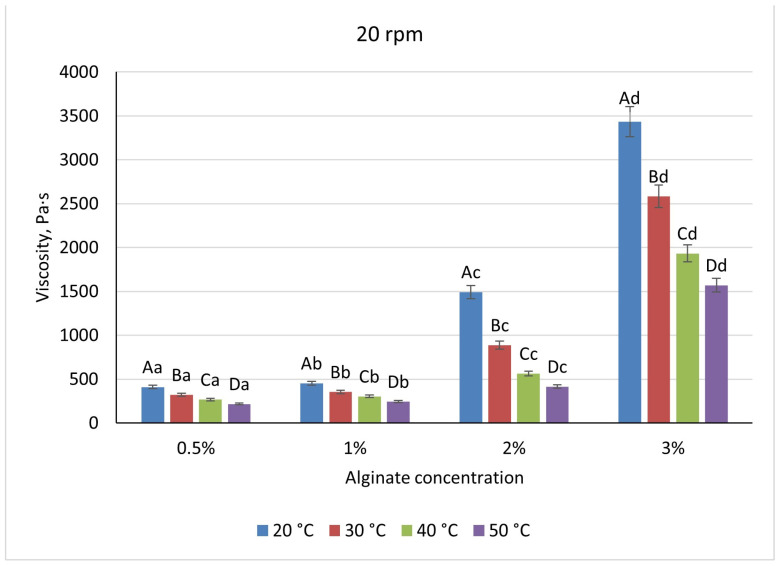
Dependence of the viscosity of the gel-forming mixture on the concentration of sodium alginate solution at different temperatures (at 20 rpm) (Different lowercase letters (a–d) indicate statistically significant differences within the same temperature (*p* < 0.05). Different uppercase letters (A–D) indicate a significant difference within the same concentration of sodium alginate (*p* < 0.05).

**Figure 6 foods-14-02878-f006:**
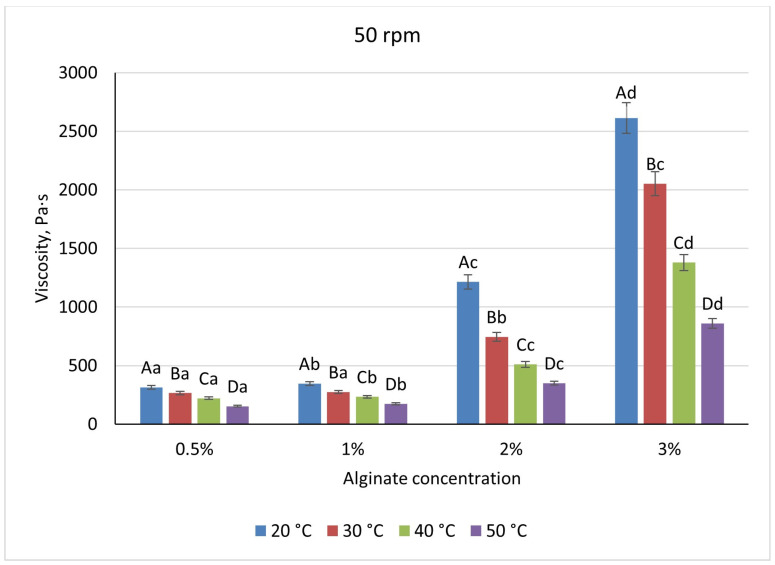
Dependence of the viscosity of the gel-forming mixture on the concentration of the sodium alginate solution at different temperatures (at 50 rpm) (Different lowercase letters (a–d) indicate statistically significant differences within the same temperature (*p* < 0.05). Different uppercase letters (A–D) indicate a significant difference within the same concentration of sodium alginate (*p* < 0.05).

**Figure 7 foods-14-02878-f007:**
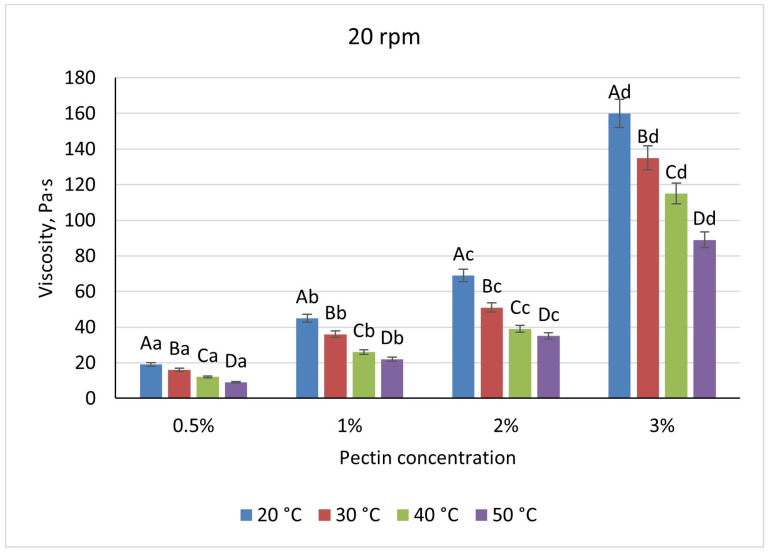
Dependence of the viscosity of the gel-forming mixture on the concentration of the pectin solution at different temperatures (at 20 rpm). (Different lowercase letters (a–d) indicate statistically significant differences within the same temperature (*p* < 0.05). Different uppercase letters (A–D) indicate a significant difference within the same concentration of pectin (*p* < 0.05).

**Figure 8 foods-14-02878-f008:**
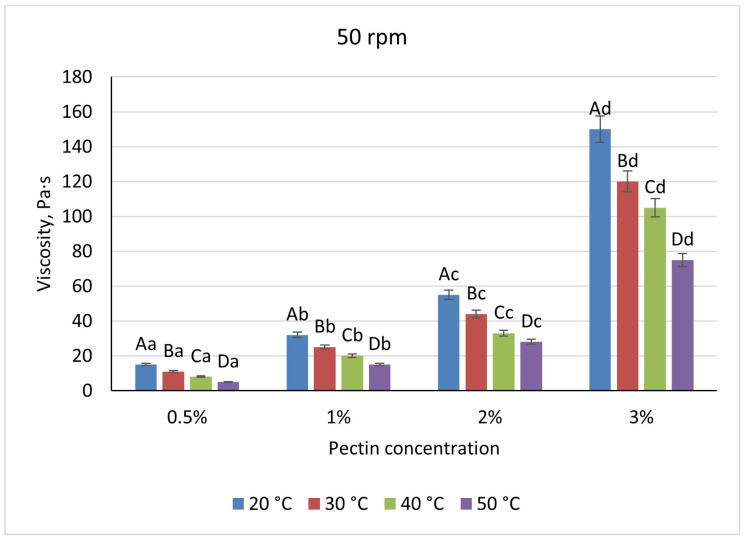
Dependence of the viscosity of the gel-forming mixture on the concentration of the pectin solution at different temperatures (at 50 rpm). (Different lowercase letters (a–d) indicate statistically significant differences within the same temperature (*p* < 0.05). Different uppercase letters (A–D) indicate a significant difference within the same concentration of pectin (*p* < 0.05).

**Figure 9 foods-14-02878-f009:**
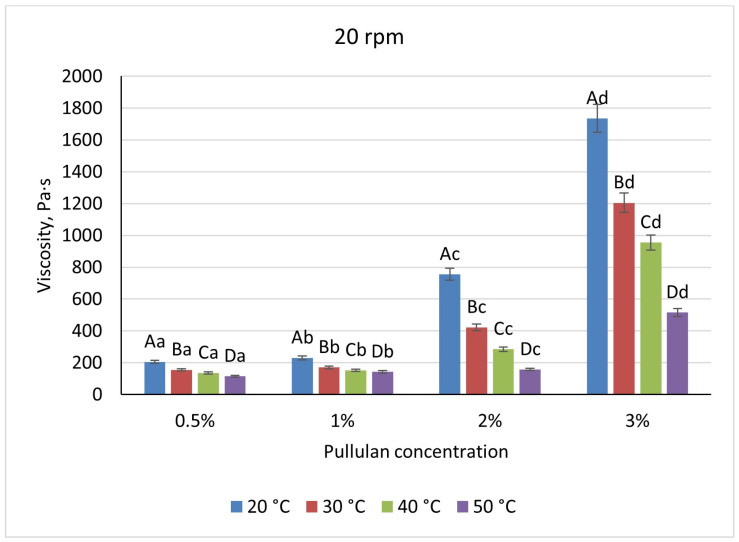
Dependence of the viscosity of the gel-forming mixture on the concentration of the pullulan solution at different temperatures (at 20 rpm). (Different lowercase letters (a–d) indicate statistically significant differences within the same temperature (*p* < 0.05). Different uppercase letters (A–D) indicate a significant difference within the same concentration of pullulan(*p* < 0.05).

**Figure 10 foods-14-02878-f010:**
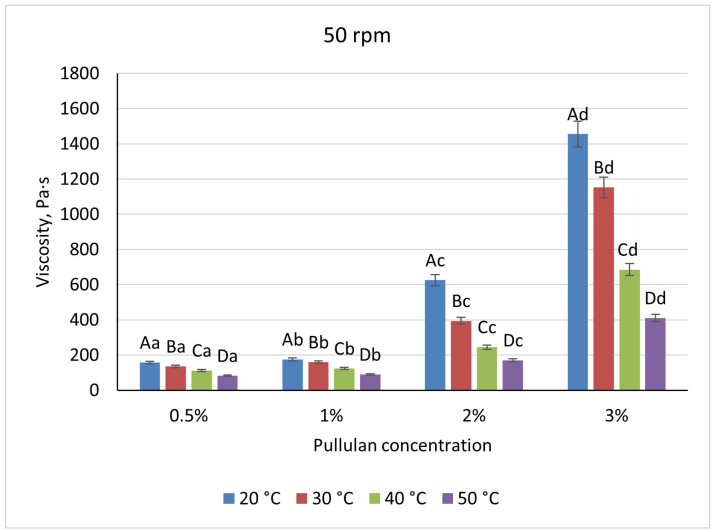
Dependence of the viscosity of the gel-forming mixture on the concentration of the pullulan solution at different temperatures (at 50 rpm). (Different lowercase letters (a–d) indicate statistically significant differences within the same temperature (*p* < 0.05). Different uppercase letters (A–D) indicate a significant difference within the same concentration of pullulan (*p* < 0.05).

**Figure 11 foods-14-02878-f011:**
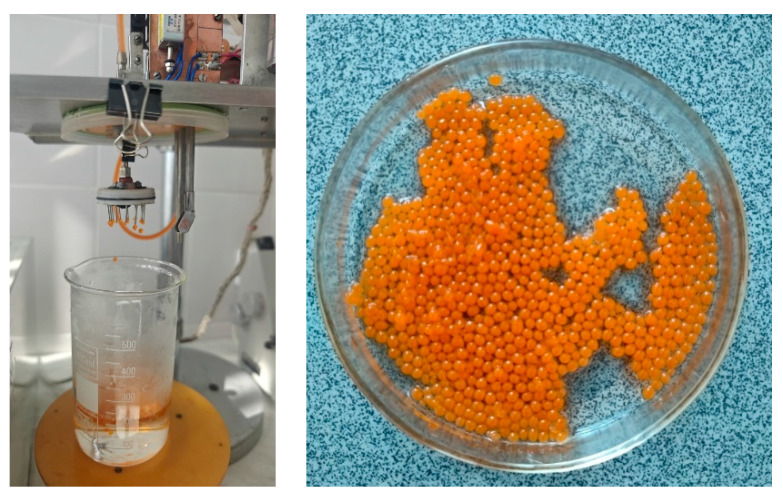
Capsules containing 1% alginate coated with pullulan (colored for visual clarity).

**Figure 12 foods-14-02878-f012:**
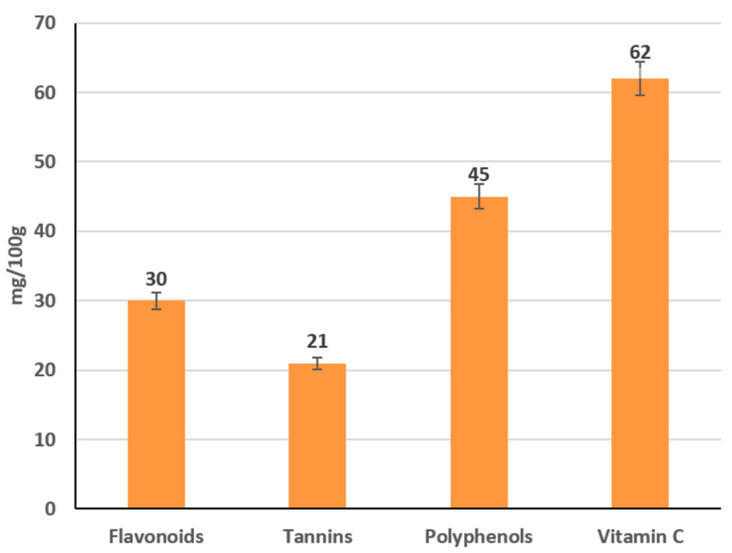
Concentrations of various phytochemicals in tinctures made from different plants.

**Figure 13 foods-14-02878-f013:**
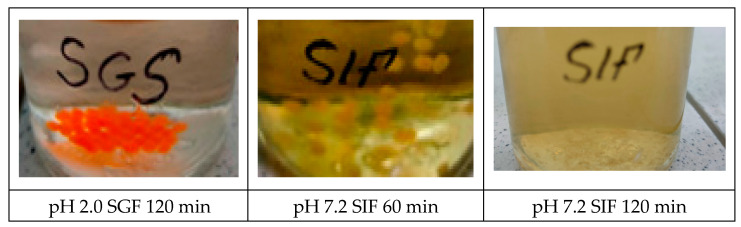
Behavior of capsules in a model environment of the gastrointestinal tract depending on time.

**Figure 14 foods-14-02878-f014:**
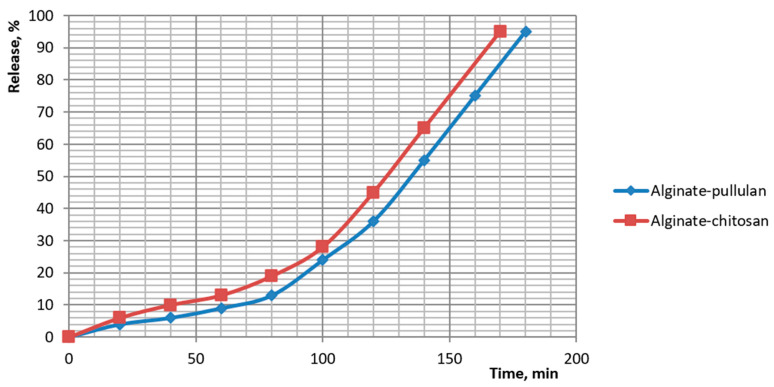
Kinetics of dietary supplement release from capsules with alginate coated with pullulan and alginate coated with chitosan.

**Figure 15 foods-14-02878-f015:**
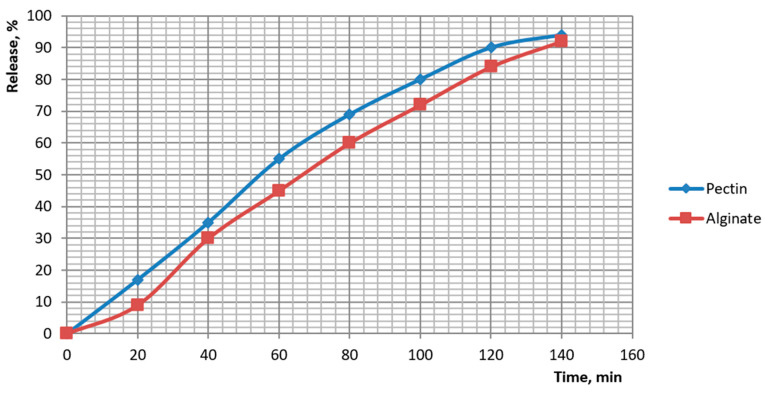
Kinetics of dietary supplement release from capsules of pectin and alginate.

**Figure 16 foods-14-02878-f016:**
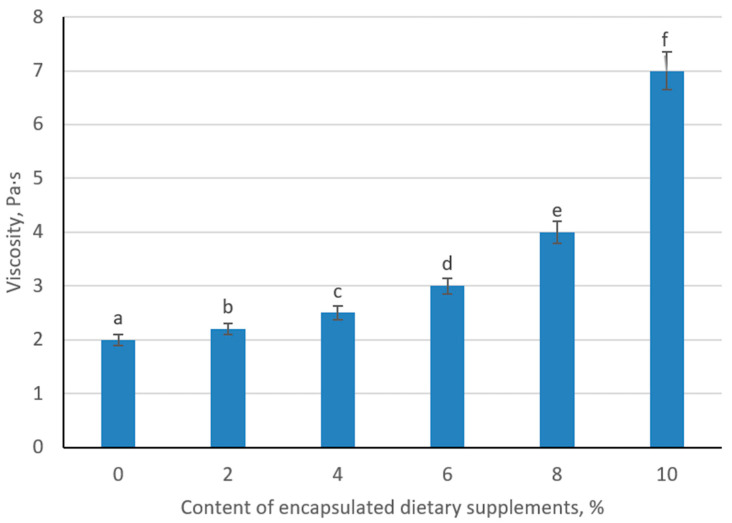
Change of viscosity of sour milk product depending on the content of encapsulated dietary supplements. (Different lowercase letters (a–f) indicate statistically significant differences between the samples (*p* < 0.05).

**Table 2 foods-14-02878-t002:** Recipe for a fermented milk product with encapsulated dietary supplements.

Ingredient	Content, %					
Milk with 2.5% fat content	95	93	91	89	87	85
Encapsulated dietary supplements	0	2	4	6	8	10
Starter culture (*Lactobacillus acidophilus*, *Streptococcus thermophilus*)	5	5	5	5	5	5

**Table 3 foods-14-02878-t003:** Physical and chemical indicators of medicinal plants.

Name of Plant	Polyphenols, %	Flavonoids, %	Hydrolyzable, Tannin, %	Condensed Tannin, %
Echinacea	1.82 ± 0.03 ^a^	1.06 ± 0.01 ^b^	16.46 ± 0.25 ^c^	3.62 ± 0.05 ^c^
Rosehip	4.47 ± 0.09 ^b^	2.75 ± 0.04 ^c^	2.70 ± 0.05 ^a^	0.83 ± 0.01 ^a^
Levzeya	5.28 ± 0.10 ^c^	0.94 ± 0.02 ^a^	4.41 ± 0.10 ^b^	2.58 ± 0.03 ^b^

^a–c^ means within the same column with different letters, differ significantly (*p* < 0.05).

**Table 5 foods-14-02878-t005:** Kinetic parameters of dietary supplement release from the capsule.

Variant	First Order		Hixson–Crowell		Korsmeyer–Peppas		Exponent **
Coefficients *	R^2^	R	R^2^	R	R^2^	R	n
Pectin	0.976	0.989	0.980	0.986	0.975	0.982	0.692
Alginate	0.972	0.980	0.988	0.992	0.982	0.989	0.690
Alginate + chitosan	0.968	0.971	0.962	0.965	0.882	0.991	0.524
Alginate + pullulan	0.985	0.991	0.948	0.976	0.989	0.992	0.511

Notes: *—coefficient of determination (R^2^), correlation coefficient (R); **—release exponent (n).

**Table 6 foods-14-02878-t006:** The effect of the amount of encapsulated dietary supplements on the sensory indicators of a fermented milk beverage with encapsulated dietary supplements.

Indicator	Control	Content of Encapsulated Dietary Supplements, %				
		2%	4%	6%	8%	10%
Smell	5	5	5	5	5	5
Taste	5	5	5	4	4	4
Color	4	5	5	5	5	5
Appearance	5	5	5	5	5	4
Consistency	5	5	5	5	5	4
Average score	4.8	5	5	4.8	4.8	4.4

**Table 7 foods-14-02878-t007:** Changes in the titratable acidity of fermented milk beverages.

Name	Storage Time, h			
	24 h	72 h	120 h	168 h
Titratable acidity, °T	87 ± 1 ^a^	98 ± 2 ^b^	119 ± 2 ^c^	130 ± 3 ^d^

^a–d^ means within the same row with different letters, differ significantly (*p* < 0.05).

**Table 8 foods-14-02878-t008:** Dynamics of changes in the organoleptic indicators of fermented milk beverages depending on storage time.

Indicator	24 h	72 h	120 h	168 h
Structure and consistency	Homogeneous, with intact clot. No gas formation		Homogeneous, with an intact clot. With slight gas formation	Heterogeneous, with strong gas formation
Taste and odor	Clean, sour milk, without any foreign tastes or odors			Excessively sour taste and smell, with a yeasty aftertaste
Color	Milky white, uniform throughout the entire mass			Milky white, uniform

**Table 9 foods-14-02878-t009:** Changes in viable counts of lactic acid microorganisms.

Sample	Lactic Acid Microorganisms, log CFU/g				
	24 h	48 h	72 h	96 h	120 h
Control	7.18	7.18	7.08	7.07	7.00
Experimental	7.18	7.18	7.11	7.08	7.04

## Data Availability

The original contributions presented in this study are included in the article. Further inquiries can be directed to the corresponding author.

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
