# Peer review of "Development of Alginate–Pullulan Capsules for Targeted Delivery of Herbal Dietary Supplements in Functional Fermented Milk Products"

_foods, 2025, doi:10.3390/foods14162878_

Round 1
Reviewer 1 Report
Comments and Suggestions for Authors
This manuscript investigates the development and optimization of alginate–pullulan capsules for the targeted delivery of herbal dietary supplements in functional fermented dairy products. Through systematic material selection, process optimization, and performance evaluation, the study confirms the feasibility of 1% alginate–pullulan capsules in terms of gastrointestinal targeted release and stability within fermented dairy matrices. The research design is clearly application-oriented, the experimental methods are relatively standardized, and the results offer valuable references for the development of functional foods. The study objectives are well-defined, the application value is prominent, the methodology is systematic and rigorous, the key results are reliable, and the references are appropriately cited. However, several issues remain that require careful revision by the authors.
- The in vitro release experiments only describe the “integrity” and “complete disintegration time” of the capsules in simulated gastrointestinal fluids, without quantifying the release kinetics of active components (such as flavonoids and tannins). It is recommended to supplement the study with release curves and kinetic model fitting (e.g., first-order kinetics, Higuchi model) to more accurately assess the efficiency of targeted delivery.
- The current discussion primarily focuses on phenomenological explanations of the experimental results and lacks horizontal comparisons with similar studies. It is suggested to include comparative analyses with relevant literature to clarify the technological advantages and limitations of this study.
- Figures 2 and 3 (microscopic images of pectin and alginate capsules) lack specific scale indicators (such as scale bars), making it difficult to visually assess particle size differences.
- All tables in the manuscript should be revised to follow the three-line table format, which is more consistent with scientific writing standards.
- The storage experiments only monitored acidity and sensory changes, without assessing the retention rates of key functional components (such as the encapsulated herbal bioactives) during storage. As a result, the study cannot directly demonstrate the long-term protective effect of the capsules on active ingredients. The authors are requested to provide an explanation or address the reasons for this limitation in the discussion.
Author Response
Dear Reviewer
Please find attached - the answer report
Thanks

Reviewer 2 Report
Comments and Suggestions for Authors
The manuscript presents a novel approach to incorporating herbal dietary supplements into fermented milk products using a spray-cut encapsulation technique. The authors provide detailed optimization of encapsulation parameters, demonstrate stability in simulated gastrointestinal conditions, and evaluate the sensory and rheological impacts of capsule inclusion. The work is methodologically sound, of potential interest to researchers in food technology and functional dairy product development, and contributes to the literature on plant extract delivery systems. However, several aspects require clarification, additional data, and improved presentation before the manuscript can be considered for publication:
- The term “spray-cut encapsulation” should be defined more precisely in the introduction. Is this a widely used term or specific to the authors’ equipment?
- While the encapsulation method is described in detail, the novelty of the spray-cut technique compared to other microencapsulation approaches (e.g., extrusion, electrostatic droplet generation, or spray-drying) is not clearly established. Please elaborate on the advantages and limitations of your method in relation to existing technologies.
- Please ensure that all microscopic or morphological images are accompanied by appropriate scale bars to allow accurate assessment of particle size and structural features.
- It remains unclear why the gastrointestinal simulation was only conducted up to the small intestinal phase. Considering that undigested materials or delayed-release systems may reach the colon, it would be valuable to either include a colonic simulation or provide a rationale for its omission.
- While the sensory quality is described as “acceptable”, details on the sensory evaluation protocol are missing. Please clarify: How many panelists participated? Was a standardized scoring method used?
- The product is described as having a 5-day shelf-life due to gas formation and sourness at 168 h. Consider providing microbiological data (e.g., lactic acid bacteria counts) to support this assessment.
- Since herbal extracts often contain natural antimicrobial compounds, their incorporation into fermented milk may affect the viability of lactic acid bacteria. The authors are advised to evaluate and report the viable counts of lactic acid bacteria before and after capsule addition to assess any potential inhibitory effects.
- Discussion section is weakly presented in the manuscript. Obtained results must be compared with other related literature.
- Please ensure consistency in the use of abbreviations and full terms throughout the manuscript (e.g., "hour" vs "h"). All units and abbreviations should be standardized according to journal guidelines.
- More attention should be paid to word spelling, space characters, singular and plurals, punctuations, the upper- and lower-case letters, subscripts, special characters, etc.
Author Response

(The authors gave the same response as above.)

Reviewer 3 Report
Comments and Suggestions for Authors
Thank you for your manuscript. Here are some comments:
- What was the dry weight of plants used in the research
- how much solution was passed through disc-spray encapsulator
- line 218 and 247- What is the unit of measurement?
- tables and pictures should be straightened, and explanations of abbreviations under the pictures should be reduced in letter size
- It is necessary to add more comparisons with other works throughout the all manuscript
Author Response

(The authors gave the same response as above.)

Round 2
Reviewer 3 Report
Comments and Suggestions for Authors
Manuscript is ready to be published